



# Statistical Reconstruction of Daily Precipitation and Temperature Fields in Switzerland back to 1864

Lucas Pfister[1,2], Stefan Brönnimann[1,2], Mikhaël Schwander[3], Francesco Alessandro Isotta[3], Pascal Horton[1,2], and Christian Rohr[1,4]

[1]Oeschger Centre for Climate Change Research, University of Bern, Bern, Switzerland
[2]Institute of Geography, University of Bern, Bern, Switzerland
[3]Federal Office of Meteorology and Climatology MeteoSwiss, Zurich, Switzerland
[4]Institute of History, University of Bern, Bern, Switzerland

**Correspondence:** Lucas Pfister (lucas.pfister@giub.unibe.ch)

**Abstract.** Spatial information on past weather contributes to better understand the processes behind day-to-day weather variability and to assess the risks arising from weather extremes. For Switzerland, daily-resolved spatial information on meteorological parameters is restricted to the period starting from 1961, whereas prior to that local station observations are the only source of daily, long-term weather data. While attempts have been made to reconstruct spatial weather patterns for certain extreme events, the task of creating a continuous spatial weather reconstruction dataset for Switzerland has so far not been addressed. Here, we aim to reconstruct daily, high-resolution precipitation and temperature fields for Switzerland back to 1864 with an analogue resampling method (ARM) using station data and a weather type classification. Analogue reconstructions are post-processed with an ensemble Kalman fitting (EnKF) approach and quantile mapping. Results suggest that the presented methods are suitable for daily precipitation and temperature reconstruction. Evaluation experiments reveal an excellent skill for temperature and a good skill for precipitation. As illustrated on the example of the avalanche winter 1887/88, these weather reconstructions have a great potential for various analyses of past weather and for climate impact modelling.

## 1 Introduction

Historical meteorological measurements are invaluable not just for studying climate variability, but also for long-term variability in weather, its extremes and its relation to the large-scale circulation. Day-to-day weather data allow the calculation of targeted indices (e.g., consecutive dry days, growing degree days), which are more useful than monthly climate data for assessing climate impacts. Moreover, daily data feed into current impact models and allow studying crop growth, water availability, or impacts of droughts, floods or avalanches numerically. However, historical station observations only capture local weather conditions. Most of the applications mentioned above require spatial fields of meteorological parameters.



For Switzerland a long-term, high-resolution and time-consistent spatial dataset of precipitation and temperature starting in 1864 is available only with monthly resolution, introduced recently by Isotta et al. (2019). A comparable daily dataset, however, which is needed to analyse past weather, covers a relatively short period starting in 1961 (MeteoSwiss, 2016a, b). Prior to 1961, observations from weather stations are the only sources that provide continuous information on daily weather. Today's dynamical and stochastic models offer new possibilities to make use of this sparse information and enable us to create

spatial reconstructions of past weather. In recent years, several efforts have been made to create high-resolution temperature and precipitation reconstructions for historical extreme events in Switzerland using dynamical (Brugnara et al., 2017; Stucki et al., 2018) and statistical (Flückiger et al., 2017) downscaling methods. While for Switzerland, the task to create long-term, high-resolution daily spatial weather reconstructions has not been addressed so far, Caillouet et al. (2016, 2019) have presented a continuous dataset of daily precipitation and temperature fields for France starting in 1871 by statistically downscaling data

from the 20CR reanalysis. This study aims at creating such a dataset of long-term, high-resolution daily spatial reconstructions of precipitation and temperature for Switzerland by extending the currently available datasets backwards in time until 1864.

    We use a statistical approach that has been applied in various research areas related to climate sciences: the so-called analogue method (Lorenz, 1969; Zorita and von Storch, 1999; Ben Daoud et al., 2016; Barnett and Preisendorfer, 1978; Kruizinga and Murphy, 1983; Horton et al., 2012). In recent years, this method has also been introduced to local-scale weather

reconstruction using historical station data (Flückiger et al., 2017; Rössler and Brönnimann, 2018). It is based on the assumption that over time similar spatial patterns of atmospheric states occur that produce similar local effects (Lorenz, 1969; Horton et al., 2017). The analogue approach makes use of this statistical relationship between large-scale and local weather or meteorological patterns, while the former is used to predict the latter. To predict a certain atmospheric feature, e.g. precipitation and temperature fields for a given day of interest, the analogue method looks for the day with the most similar predictor values (best analogue)

and takes the atmospheric feature from this (or multiple) best analogue day(s) as prediction (Zorita et al., 1995). As it is basically a resampling of observed states of the atmosphere (spatial weather data) along the time axis to optimally fit certain predictors (Graham et al., 2007; Franke et al., 2011), the term analogue resampling method (ARM) is used in this paper.

    In our approach, analogue reconstructions are further improved. Using techniques borrowed from data assimilation, reconstructed temperature fields are adjusted towards station measurements with a so-called ensemble Kalman fitting approach

(Whitaker and Hamill, 2002; Franke et al., 2017) that is adapted to analogue reconstructions. Reconstructed precipitation data are bias-corrected using a quantile mapping method (Gudmundsson et al., 2012) by fitting reconstructed to observed precipitation distributions.

    The result is a long-term, daily-resolved spatial dataset of precipitation and temperature with a 2.2×2.2 km spatial resolution for the period of 1 Jan 1864–31 Dec 2017. Reconstructions are evaluated against gridded data from MeteoSwiss and against

station observations. To demonstrate the potential of the reconstructions, we analyse the avalanche winter in 1887/88, comparing reconstructions to previous studies, as well as documentary data (Vieli, 2017; Coaz, 1889). This paper accompanies the online publication of the reconstructed precipitation and temperature datasets at the open-source repository PANGAEA (https://doi.org/10.1594/PANGAEA.907579).





The paper is organised as follows: Section 2 provides an overview of the data used. Section 3 describes the methods of weather reconstruction and post-processing and presents validation strategy and –measures applied for assessing the reconstructions. In Section 4, results from the validation of reconstructed and post-processed temperature and precipitation fields are presented and discussed before analysing the avalanche winter 1887/88 in section 5. Conclusions are drawn in Section 6.

## 2  Data

Statistical weather reconstruction methods require long-term and if possible homogeneous series of station measurements. In Switzerland, we can benefit from the network of MeteoSwiss going back to the year 1864 (Füllemann et al., 2011; Begert et al., 2005). All 68 meteorological stations used for reconstruction are part of the Swiss National Basic Climatological Network (Swiss NBCN), a network of long-term, continuous and high-quality measurements used for climate monitoring (Begert et al., 2007; Begert, 2008). To ensure the consistency of the reconstructions over time, the set of meteorological stations and parameters used is ideally not changed over time. Therefore, only measurement series starting prior to 1901 and continuing until today with interruptions of no longer than five years were selected. One exception is the station of Grand St-Bernard (GSB), where data show a gap from 30 Jul 1925 to 31 Dec 1933. This station was included, as it lies within a data-scarce region and is representing higher altitudes. In all cases, homogenised daily mean temperature and precipitation sums were used, as well as daily mean pressure values at station height (QFE). An overview of measurement locations and variables, as well as their vertical distribution is given in Fig. 1.

In total we used 10 pressure, 25 temperature and 67 precipitation series. The large number of precipitation measurements was chosen to account for high spatial variability of this variable, while 10 stations are enough to cover surface pressure. Most stations are located at lower elevations and in valleys, while higher altitudes (hillsides, mountains) are under-represented. Two independent measurement series (yellow) from the Swiss Plateau and the Alpine region were used for validation: Schaffhausen (SHA, 438 m a.s.l.) with measurements from 1 Jan 1864–31 Dec 2017 and Grimsel-Hospiz (GRH, 1980 m a.s.l.), covering the period from 1 Jan 1932–31 Dec 2017. Note that data from Schaffhausen are not homogenised.

Furthermore, the ARM requires spatial data, from where best analogue reconstructions are drawn. Here we used daily gridded precipitation and temperature data provided by MeteoSwiss (MeteoSwiss, 2016a, b) with a spatial resolution of 2.2 km, covering the period 1 Jan 1961–31 Dec 2017. Precipitation data (RhiresD) indicate accumulated precipitation (rain- and snowfall water equivalent) from 06:00 UTC to 06:00 UTC of the following day (MeteoSwiss, 2016a), spatially interpolated from daily precipitation sums measured at the MeteoSwiss high-resolution rain-gauge station network. Topographic effects and differences in station distribution are accounted for. Errors are estimated to be in the order of factor 1.7 for precipitation below the 20% quantile (tendency towards overestimation) and 1.3 for precipitation above the 90% quantile (tendency towards underestimation) and are higher in mountainous areas (MeteoSwiss, 2016a). A detailed description of this dataset and the methods to derive it can be found in MeteoSwiss (2016a) and Schwarb (2001).

Gridded temperature (TabsD) displays daily mean (00:00 to 00:00 UTC) air temperature measured in degrees Celsius at 2 m above ground (MeteoSwiss, 2016b). As homogenized station data were used for interpolation, errors resulting from changes





of measurement location or instruments are corrected. Regional differences in vertical temperature gradients, as well as the effects of warm boundary layers and temperature inversions are taken into account. Standard errors in the TabsD dataset range from 0.6 to 1.1 °C in the Swiss Plateau (smaller in summer) and reach values of 4 °C in inner Alpine valleys in winter. For

further information on interpolation method and validation, the reader is referred to Frei (2014) and MeteoSwiss (2016b).

Furthermore, a daily weather type (WT) classification is used (Schwander et al., 2017), covering the period of interest. These WT reconstructions are based on the CAP9 classification used by MeteoSwiss that distinguishes 9 different WTs for Central Europe (Weusthoff, 2011), which show good skills at predicting daily weather, especially precipitation in the Alpine region (Schiemann and Frei, 2010). Merging two pairs of similar CAP9 WTs, Schwander et al. (2017) reconstructed WTs from 1763

to 2009 using homogenised instrumental measurement series from different locations in Europe. For each day, this dataset provides the probabilities of each CAP7 WT. WTs from 2010 onwards were calculated from the CAP9 data from MeteoSwiss (Weusthoff, 2011).

As argued by Schwander et al. (2017), reconstructed WTs are more reliable for winter than for summer months, as the underlying meteorological patterns are more pronounced during winter. For weather reconstruction, this property has to be

taken into account.

## 3 Methods

### 3.1 The Analogue Resampling Method

The application of the ARM in this paper is based on the work by Flückiger et al. (2017). The ARM requires two meteorological archives: data to predict the spatial fields and a record of the spatial data from which the reconstructions are drawn. To predict

the spatial fields we used daily station observations, while the RhiresD and TabsD datasets for 1961–2017 from MeteoSwiss serve as record of spatial data (see chapter 2). For a given day in the past, we screen the period for which we have spatial data (analogue pool) for the most similar day in terms of station data (best analogue). Precipitation and temperature fields from this day serve as an estimate for the day in the past.

The ARM has the advantage of preserving natural variability and spatial patterns in the reconstructions (Zorita and von

Storch, 1999). With input from both, coarser-resolved data, e.g. reanalyses or weather types, as well as local information (station data), it can make use of more data sources than simple downscaling or interpolation of station observations alone. A limitation is the size of the analogue pool, which has to be large enough to provide reasonably matching analogues to a given atmospheric state (Zorita and von Storch, 1999). Furthermore, temporal consistency is not guaranteed.

In order to maintain the physical consistency of the reconstructions, further conditions are established:

1) The day of interest and possible analogue days are required to be of the same WT to assure similar synoptic-scale atmospheric conditions, e.g. wind fields (Weusthoff, 2011). To account for the uncertainty in WT reconstructions, we did not restrict the analogue to the most probable WT but accepted additional WTs such that they cover the true WT with a combined probability of at least 95% according to Schwander et al. (2017).





2) The day of interest and possible analogue days are required to be within the same season to account for seasonally
   different spatial patterns. The time window is set to ±60 days centred at the target day.

Following these conditions, the best analogue is defined as a day within the analogue pool with the same weather type, within the same time window that shows the most similar values of certain meteorological variables from a defined set of stations to the day of interest.

Before the application of the ARM, station and gridded data are pre-processed. As observed variables have different scales, each measurement series is standardised. Temperature data are decomposed into a smoothed mean climatology and the respective anomalies. For each observation series, as well as each cell of the gridded data, a smoothed mean seasonality curve is estimated by fitting the first two harmonics of temperature time series following equation 1, using linear regression.

$$S = c_0 + c_1 \sin\left(\frac{2\pi doy}{n_{doy}}\right) + c_2 \cos\left(\frac{2\pi doy}{n_{doy}}\right) + c_3 \sin\left(\frac{4\pi doy}{n_{doy}}\right) + c_4 \cos\left(\frac{4\pi doy}{n_{doy}}\right) \tag{1}$$

$doy$ denotes the day of year, $n_{doy}$ the number of days in a year and $c_0$, $c_1$, $c_2$, $c_3$ and $c_4$ are parameters to estimate. After the calculation of the analogue reconstructions using temperature anomalies, mean climatology is then again added to the reconstructed temperature deviation fields to get absolute temperature data. This procedure slightly alters the characteristics of the ARM, as adding climatology and resampled temperature deviations does not only resample known temperature fields, but creates new ones. An elimination of the signal from climatic temperature changes over the last centuries did not improve the results and was not further pursued in this study.

With pre-processed data, the analogue method is applied. Following Horton et al. (2017), the root mean squared error (RMSE) is used as measure of similarity (equation 2).

$$d(x,y) = \sqrt{\sum_{i=1}^{n}(x_i - y_i)^2} \tag{2}$$

where $x$ and $y$ are vectors of observations from the day of interest and a day within the analogue pool, respectively. $i$ denotes the different observations within this vector. Other measures of similarity like the Mahalanobis distance were not examined.

## 3.2 Post-Processing Methods

The best analogue may not perfectly fit all observations. To further improve the temperature reconstructions, we borrow from data assimilation techniques (see e.g. Daley, 1999; Kalnay, 2007). The method used here is based on the ensemble Kalman filter (Kalman, 1960; Evensen, 1994; Burgers et al., 1998), which is applied e.g. for data assimilation of ensemble forecasts from dynamical models. Here we use the best analogue in the same way as forecast (termed background or first-guess) and the best $n$ analogues as ensemble. However, neither the analysis nor the covariance matrix (see below) are passed on to the next time step. This simplification is called ensemble Kalman fitting (EnKF) (Bhend et al., 2012; Franke et al., 2017) or off-line data assimilation (Matsikaris et al., 2015). The EnKF essentially minimizes a least-squares problem (Franke et al., 2017). The state vector $x$ that minimizes the following cost function $J$ is optimal in the case of Gaussian errors:

$$J(x) = \left(x - x^b\right)^T \left(P^b\right)^{-1}\left(x - x^b\right) + \left(y - H[x]\right)^T R^{-1}\left(y - H[x]\right) \tag{3}$$





where $x^b$ is the first guess (background), in this case reconstructions from the best analogue. $P^b$ is the background error
covariance matrix that in this particular case is estimated from an "ensemble" of the best $n$ analogues, i.e. temperature fields
from the $n$ most similar days to the day of interest. Vector $y$ contains station observations and operator $H$ is used to extract the
observations from the model space. $R$ is the error covariance matrix of $y - H(x)$. With station observations and the ensemble
of the best $n$ analogues, a new estimation of temperature $x^a$ that is the best estimate for true atmospheric state $x$ is calculated
from equation 4

$$x^a = x^b + K(y - Hx^b) \tag{4}$$

where $x^a$ denotes the updated state vector (analysis), $x^b$ and $y$ as described above and $K$ is the Kalman gain or innovation
matrix calculated from the ensemble. In this and the following equations, $H$ describes the Jacobian matrix of $H(x)$ and extracts
the values from the grid cell closest to the observation site of $y$.

We use an implementation (Whitaker and Hamill, 2002; Bhend et al., 2012) in which each observation is assimilated se-
quentially (equations 5–7). The fitting procedure is split into two steps: an update of the ensemble mean $\bar{x}$ (equation 5a) and
an update of the anomalies $x'$ with respect to the ensemble mean (equation 5b). Equations 6a and 6b depict the calculation of
the Kalman gain $K$ for the ensemble mean and $\tilde{K}$ for the anomalies.

$$\bar{x}^a = \bar{x}^b + K(\bar{y} - H\bar{x}^b) \tag{5a}$$

$$x'^a = x'^b + \tilde{K}(y' - Hx'^b) \text{ with: } y' = 0 \tag{5b}$$

$$K = P^b H^T (H P^b H^T + R)^{-1} \tag{6a}$$

$$\tilde{K} = P^b H^T \left[ \left( \sqrt{H P^b H^T + R} \right)^{-1} \right]^T \times \left( \sqrt{H P^b H^T + R} + \sqrt{R} \right)^{-1} \tag{6b}$$

$\bar{x}^a$ and $\bar{x}^b$ denote the analysis and background of the ensemble mean and $x'^a$ and $x'^b$ the corresponding anomalies. $P^b$ and
$R$ are the error covariance matrices as in equation 3. The observation error $R$ is roughly estimated to be 1 °C. The background
error covariance matrix $P^b$ is calculated from the best $n$ analogues following equation 7, where $i$ and $j$ denote grid boxes and
$k$ the ensemble members.

$$P^b_{i,j} = \frac{1}{n_{ens} - 1} \sum_{k=1}^{n_{ens}} \left( x^b_{i,k} - \bar{x}^b_k \right) \left( x^b_{j,k} - \bar{x}^b_k \right) \tag{7}$$

Following Whitaker and Hamill (2002), not the full error covariance matrix is calculated, but directly the conversion $P^b H^T$
in order to save computational resources. Covariance matrices estimated from small samples may exhibit spurious covariances




far away from the observation. Spatial localisation is often used to minimise these effects. In our case, the study areas is too
small and the ensemble size sufficient such that localisation is not necessary (tests using a Gaussian weighting function did not
show improvement).

For each day, the EnKF is applied to the analogue reconstructions using a selection of measurement series that exhibit
an average monthly correlation with co-located data from TabsD above 0.975 (see Fig. 1). This is to avoid measurement
series subject to local influences, which are not resolved by spatial data and thus would lead to erroneous assimilations. To
account for a bias between local measured temperature at a weather station and spatially aggregated temperature values of the
corresponding grid cell, station data are corrected by subtracting the mean bias between measurement and grid cell value from
the TabsD dataset over the period 1961–2017 for each month. This procedure prevents systematic biases in fitted temperature
fields. The ensemble size is set to the 50 best analogues.

Precipitation reconstructions are often affected by biases in the mean, an increased number of wet days and underestimation
of extreme events (Piani et al., 2010b). To avoid such effects, analogue reconstructions of precipitation are post-processed.
Although attempts have been made to assimilate precipitation with an application of a Kalman filter (Lien et al., 2013, 2016),
in this paper, a much simpler approach is used: quantile mapping (QM). This method of model output statistics aims at trans-
forming cumulative distribution functions (CDF) of modelled precipitation to match the CDFs of observed precipitation by
finding a statistical transfer function $h$ (Maraun et al., 2010; Maraun, 2013), i.e. it is mapping modelled to the observed distri-
bution. As pointed out by Cannon (2018), this procedure is asynchronous, that is not considering any chronological aspects of
precipitation. In its simplest application, QM corrects the model bias according to observed precipitation values (Piani et al.,
2010a) and can be generally expressed by equation 8.

$$P_o = h(P_m) \tag{8}$$

where $P_o$ and $P_m$ are observed and modelled precipitation, respectively and $h$ the transfer function (Gudmundsson et al.,
2012). Based on the probability integral transform theorem (Angus, 1994), the transformation can be described as:

$$P_o = F_o^{-1}(F_m(P_m)) \tag{9}$$

with $F_m$ the CDF of modelled precipitation and $F_o^{-1}$ the inverse CDF of the observed precipitation. To solve this equation,
the distribution of the variable of interest has to be defined. In this paper, a parametric transformation using an exponential
asymptotic function to estimate precipitation distribution was chosen following Gudmundsson et al. (2012). This parametric
transformation is described by equation 10, where $\hat{P}_o$ denotes the best estimate of $P_o$ and parameters $a$, $b$, $x$ and $\tau$ are to be
determined.

$$\hat{P}_o = (a + bP_m)\left(1 - e^{-(P_m-x)/\tau}\right) \tag{10}$$

The best prediction of parameters $a$, $b$, $x$ and $\tau$ is estimated by minimising the residual sum of squares for wet days (Gud-
mundsson et al., 2012). To define wet days, a threshold for $P > 0.1$ mm was set. Precipitation values beyond 0.1 mm were
set to zero. Parametric transfer functions were calculated from all data within the calibration period 1 Jan 1961–31 Dec 2017





for each grid cell after Piani et al. (2010a) with $P_m$ the values from the analogue reconstructions and $P_o$ the values from the RhiresD dataset. No discrimination between different seasons has been made. Based on the assumption, that the transfer function derived from this period is robust, i.e. precipitation distribution is not subject to changes in time, these functions can then be extrapolated in time to transform precipitation distributions of the reconstructed datasets back to 1864.

Note that the method as applied in this paper only corrects model bias. This simple application of QM was chosen to be

able to extrapolate distribution correction in time, as more complex approaches would likely be less robust. To substantially improve e.g. dry/wet day discrimination or extreme values, other approaches have to be applied (Cannon et al., 2015).

### 3.3  Validation

The validation of precipitation and temperature reconstructions is following common measures and strategies used in validation of field forecasts (Wilks, 2009; Jolliffe and Stephenson, 2012). If not indicated otherwise, validation measures and skill scores

are computed on absolute values.

We use the Pearson correlation coefficient for temperature, while for non-Gaussian distributed precipitation the Spearman correlation is calculated. Note that for temperature, correlation is computed on anomalies from mean seasonality (compare chapter 3.1) so it reflects day-to-day variability rather than the seasonal cycle. Error magnitudes are indicated as root mean squared error (RMSE), as this measure is sensitive to larger errors. Furthermore, systematic biases between reconstructions

and observations are evaluated.

Additionally, the mean squared error skill score (MSESS) or reduction of error-statistic (RE-value) is calculated for temperature reconstructions following equation 11, allowing us to analyse the skill of reconstructions compared to mean climatology in terms of the mean squared error.

$$MSESS = 1 - \frac{\sum\limits_{i=1}^{n} \left(x_i^{rec} - x_i^{ref}\right)^2}{\sum\limits_{i=1}^{n} \left(x_i^{0} - x_i^{ref}\right)^2} \tag{11}$$

with $x^{rec}$ the reconstruction, $x^0$ a 'no knowledge prediction' (in this case mean climatology), $x^{ref}$ the reference data from TabsD and $i$ denotes time step (validation over time) or grid cell (validation over space). A MSESS value of 1 indicates a perfect reconstruction. With an MSESS of zero, prediction skills of reconstruction and climatology are equal and values below zero denote a decline in skill compared to climatology (Jolliffe and Stephenson, 2012). Note that this measure punishes variance, i.e., a reconstruction with the correct variance but zero correlation will have an MSESS of -1.

For precipitation reconstructions, another property of interest is the discrimination between wet and dry. For this purpose, the Brier score (BS) was calculated (equation 12) that compares the predicted probability of an event to observations (Wilks, 2009).

$$BS = \frac{1}{n} \sum\limits_{i=1}^{n} (y_i - o_i)^2 \tag{12}$$





where $y$ and $o$ denote the probability of rain in reconstructions and observation, respectively and $i$ as above. As reconstruc-
tions do not provide probabilities, $y$ and $o$ are binary with 1 = rain and 0 = no rain with a wet/dry-threshold of 0.1 mm. The BS
describes the percentage of time steps (or grid cells) that was wrongly assigned as wet or dry.

In a first part, a leave-one-out validation was performed on daily gridded data within the period 1 Jan 1961–31 Dec 2017.
For each day, the best analogue day is calculated, excluding data from 5 days before and after the day of interest, as spatial
patterns from neighbouring days can be similar. Precipitation and temperature reconstructions are then validated against the
RhiresD and TabsD dataset, respectively. To analyse the full timespan of the dataset, reconstructions are compared to station
observations in a second part. For this purpose, two independent station series from Schaffhausen and Grimsel-Hospiz (see
chapter 2) were used. Measurements were compared to reconstructions by extracting values from the corresponding grid cells
without interpolation.

## 4 Results and Discussion

As described in chapter 3.3, a leave-one-out validation over the period 1961–2017 was performed and reconstructions were
compared to the MeteoSwiss RhiresD and TabsD datasets, as well as station data. In this section, we will illustrate and discuss
general results from grid-based validation of precipitation and temperature reconstructions for 1961–2017. Of particular interest
are seasonal differences and extreme events, where we evaluate also the accuracy of reconstructions to reproduce spatial
patterns. Furthermore, we compare reconstructed time series for Schaffhausen and Grimsel-Hospiz to corresponding station
observations.

### 4.1 Leave-one-out Validation in Time

Figure 2 shows results from the validation over time for analogue precipitation reconstructions (a–d) and quantile-mapped
data (e–h) against RhiresD data. Depicted are rank correlation (a, e), RMSE (b, f), mean bias (c, g) and Brier score (d, h).
The Spearman correlation coefficient for analogue reconstructions is 0.79 on average and attains values from 0.62 to 0.86
with maximum values in central Switzerland (a). Quantile mapping does not change the ranks of precipitation distribution,
therefore the two correlation maps are identical. Regarding the RMSE (e, f), an average error magnitude of less than 5 mm
in the Swiss Plateau, as well as the inner-alpine valleys and large parts of the canton of Grisons can be observed. Errors are
larger in mountainous areas and in Ticino reaching values of 6–15 mm. Post-processed data (f) reveal a negligible increase of
these errors in the range of 0.1–0.6 mm. Analogue reconstructions show a negative bias between 0.2 and 0.5 mm in the Swiss
Plateau (c). The underestimation is more pronounced in mountainous regions and in Ticino with values of 0.5–1.6 mm. Using
the quantile mapping approach described in chapter 3.2, this bias could be eliminated for the given timespan (g). The Brier
score indicates relatively high error rates in the discrimination between wet and dry days at individual locations with values
between 0.13 and 0.23 (d). Post-processed data reveal slightly negative changes in terms of Brier scores (h).

While rank correlation values show satisfying results, bias and RMSE patterns of ARM reconstructions could possibly be
explained by an underestimation of extreme and convective precipitation, which occur in the Alpine region and in Southern





Switzerland, especially in summer. While quantile-mapped data correct the bias, error values still remain large. We will elaborate on this issue below, where we look at seasonal patterns and extremes. Another problem of the reconstructions is indicated by the Brier score: on average, 17% of days are wrongly assigned as wet or dry. This relatively high fraction is not improved with post-processing, as the quantile mapping approach used here is not designed to address this particular problem.

Validation of temperature reconstructions over time in Fig. 3 reveals a good correlation already for unprocessed data, ranging between 0.76 and 0.95 with a mean of 0.91 (a). Correlation is slightly lower in Ticino and the southern valleys of Grisons. With ensemble Kalman fitting, Pearson correlation could be increased to values between 0.83 and 0.99 and a mean of 0.96, showing similar spatial patterns (e). Also the error (RMSE) could be reduced with post-processing from 1.52 °C to 0.96 °C on average (b, f). In the Swiss Plateau, the error attains values below 1 °C, while in the Alpine region, in the Jura Mountains

and in southern Switzerland RMSE values up to 2.7 °C can be observed. Unprocessed reconstructions show a systematic overestimation of temperature in the Swiss Plateau, in the Rhone valley in Valais and in the northern valleys of Ticino with values up to 0.06 °C (c). On the other hand, temperatures at higher altitudes and in southern Ticino are underestimated by 0.05 to 0.15 °C. Post-processed data (g) show less bias and a more balanced spatial pattern with values ranging between –0.08 °C and 0.03 °C and a mean of –0.01 °C. The MSESS compared to mean seasonality (d, h) is high all over Switzerland and could

be increased from 0.83 to 0.93 on average using EnKF. The pattern is following correlation.

Overall, reconstructed temperature fields can be considered to very accurately reproduce the temporal evolution of the weather. Errors are relatively low, although in regions with sparse meteorological observations, larger errors are observed. Station coverage thus plays a crucial role for analogue reconstructions. The local field of larger errors in the western Jura near La Brévine might be explained by cold air pooling, which occurs frequently in this region during winter (Vitasse et al., 2017)

and is not captured by any of the measurement series used for reconstruction. Bias patterns suggest that ARM reconstructions have problems to correctly reproduce vertical temperature gradients. A major issue here could also be inversions. In the vertical distribution of used temperature stations higher altitudes are only sparsely covered (see Fig. 1), making the correct determination of vertical gradients and inversion heights difficult. Post-processed data seem to solve large parts of this problem, but this needs for further investigation.

To find possible explanations behind the issues mentioned above and to gain more insight into the dataset, precipitation and temperature reconstructions are assessed in detail for differences between seasons, as well as extremes.

Figure 4 depicts rank correlation (a–d), RMSE (e–h) and averaged bias (i–l) over time of post-processed precipitation for each season. We see a relatively uniform correlation pattern over all seasons with slightly higher values for summer (JJA) along the northern Prealps (c). Correlation values are lowest on the southern side of the Alps, especially in winter (a). Error values

(e–h) show a similar spatial pattern throughout the year and are smallest from December to February (DJF) and slightly higher in spring (MAM) and autumn (SON); maximum values of the RMSE occur during the summer months and reach values of 8–15 mm in Ticino. Mean bias over time (i–l) still shows minor seasonal differences. In the Swiss Plateau, the mean deviation is approximately zero, except for a slight positive bias in summer. In the Jura Mountains, a minor underestimation in winter and an overestimation of summer precipitation is observed. Largest differences occur in the Alpine region and in southern





Switzerland, where winter and summer precipitation show a tendency towards overestimation, while in spring post-processed precipitation fields exhibit a negative bias in western Valais and the Gotthard region.

The pattern of RMSE with higher values during the warmer periods of the year and maxima in summer is supporting the previous assumption that reconstructions have problems to reproduce intensive or convective precipitation. Especially the latter which are local-scale phenomena may not be detected by measurement stations, making station coverage again an important 310 issue to obtain reliable reconstructions.

Analysing the same for temperature (Fig. 5), we see that Pearson correlation values (a–d) exhibit maximum correlation values in spring and summer, while in autumn and winter these values are slightly lower. The RMSE (e–h) is higher in winter than during the other seasons and reaches minima in the summer months. Maximum errors of up to 3 °C occur during winter in the Alpine region and the Jura Mountains (e). Overall, average bias (i–l) is only marginal for all seasons with values between 315 –0.2 °C and 0.1 °C. Generally, vertical temperature gradients seem to be corrected by Kalman fitting. However, we can see higher values and a distinct spatial pattern related to topography in winter, as can also be seen in the RMSE (e). This indicates that inversions, which occur more frequently during this season, remain a problem also in post-processed reconstructions.

The above-mentioned issue with cold air pools in the western Jura seems to be confirmed by the seasonal error patterns, although larger errors for this region persist throughout the year. Interestingly, the mean bias shows an underestimation of 320 winter temperature for this region.

## 4.2 Leave-one-out Validation in Space

Post-processed temperature and precipitation data were further assessed by quantiles of Swiss mean temperatures and precipitation 1961–2017 calculated from the MeteoSwiss TabsD and RhiresD datasets to analyse the accuracy of reconstructions in reproducing extremes (Fig. 6). Note that, as quantiles were calculated for average values over Switzerland, local extremes do 325 not necessarily correspond to highest or lowest quantiles for the whole area. In the following, results from validation over space are shown to analyse the capability of reconstruction methods to reproduce spatial patterns. For comparison, validation results from the analogue method are indicated in grey.

Spatial correlation of precipitation (top) is relatively low for low to moderate precipitation events and increases with precipitation quantiles. Looking at the RMSE, mean and also the spread of errors increase with increasing precipitation. While the 330 RMSE shows a median of less than 5 mm up to the 70% quantile, for extreme precipitation events above the 95% quantile errors attain values of 10–15 mm in the interquartile range. Compared to unprocessed data, a slight decrease of correlation and increase of the RMSE is visible. However, the bias is considerably improved. While analogue reconstructions reveal a strong underestimation of extreme events, the median bias becomes approximately zero for all quantiles. Uncertainties, however, remain large. The Brier score reveals that for days with zero to low precipitation as well as for extreme events, the precipitation 335 area is well represented in the reconstructions. Problems here lie in the correct reconstruction of precipitation areas for moderate events. Compared to unprocessed data, quantile mapping leads to a slightly better discrimination between wet and dry grid boxes for upper quantiles, while the BS becomes larger for lower quantiles.





From this, we can conclude, that reconstructions provide accurate precipitation fields for low to moderate precipitation events. For the benefit of unbiased reconstructions, a slight decrease of correlation and an increase of the RMSE and BS have to be accepted. Extreme events, however, are underestimated by ARM reconstructions and show large errors also for post-processed data. As extreme events by definition occur more rarely, the number of suitable analogues is limited. As argued in chapter 3, a bigger size of the analogue pool would lead to more accurate results also for extremes. Different post-processing methods might help to improve reconstructions, especially regarding wet/dry discrimination and extremes.

Validation of temperature by spatial mean temperature quantiles (bottom) shows a considerable improvement for post-processed data, compared to analogue reconstructions. Correlation values exhibit slightly better correlations for extreme temperatures, while reconstructed fields for medium temperatures are less correlated with TabsD data. RMSE values are higher for upper and lower extreme values. In general, errors could be significantly reduced with Kalman fitting. The average bias reveals, that while analogue reconstructions tend to overestimate negative extreme values and underestimate extremely high temperatures, post-processed temperature data show a median of approximately zero for all quantiles. The bias pattern of the ARM can be explained as for precipitation by a limited number of suitable analogues for extreme events. Kalman fitting solves this problem. Furthermore, the spread of bias values is within ±1 °C for four times the interquartile range. Post-processed temperature reconstructions are thus accurate and precise also for extreme temperatures. MSESS values are better for upper and lower quantiles and show worse results around the median. As days around median temperature are closer to average climatology, this pattern has to be expected. Nonetheless, the MSESS of post-processed data is still within the area of natural variability (see chapter 3.3).

### 4.3 Validation Against Independent Observations

In Fig. 7, reconstructed precipitation and temperature is compared to station observations from Schaffhausen (left) and Grimsel-Hospiz (right) over the full length of the respective series. Plotting reconstructed values against observations, we see a large spread of values for analogue reconstructions (grey), as well as for post-processed data (red). While for Schaffhausen, quantile-mapped precipitation exhibits a distribution closer to station observations in the QQ-plot (centre) compared to ARM reconstructions, we can see a tendency towards overestimation in post-processed precipitation reconstructions for Grimsel-Hospiz. The seasonal pattern (bottom) reveals lowest differences in autumn, whereas the uncertainty is highest during summer, which is again in line with more frequent convective activity during the latter season as discussed before. The systematic underestimation of precipitation by the ARM is adjusted by post-processing for all seasons. The larger uncertainty in the Alpine region discussed before is also visible in station data where the spread of bias values over all seasons (bottom) is larger for Grimsel-Hospiz. This is also the case for the RhiresD gridded dataset used for reconstruction (see chapter 2). The differences between reconstructions and station observations might at least partly be explained by the high spatial variability of precipitation, thus spatially coarser gridded data can differ considerably from local measurements. For reasons of higher spatial variability, a less perfect fit has to be expected compared to temperature reconstructions. These are closer to observed values compared to precipitation and show a smaller spread of deviations. Kalman-fitted reconstructions are even more precise; not only that the spread of values is reduced, but also the tilt in distribution could largely be corrected. Seasonal patterns of ARM reconstructions show





larger deviations during spring and autumn. These seasonal differences are eliminated by EnKF. Reconstructed temperature fields thus accurately reproduce local temperature measurements, even for remote locations. With precipitation data, however, one has to be more careful when generating station series at individual locations.

### 375    4.4    The Avalanche Winter of 1887/88

The winter of 1887/88 was one of the most severe avalanche winters during the last 150 years, boosting the efforts in avalanche prevention in Switzerland (SLF, 2000; Margreth, 2019). Intensive snowfall in February and March 1888 brought large snow masses to Switzerland leading to 1094 disastrous avalanches, damaging 850 buildings and destroying over 1300 hectares of forest and burying 49 people under the snow (Coaz, 1889; Laternser and Pfister, 1997). Documentary data from Coaz (1889)
provide a detailed description of this winter and comprehensive survey of avalanche activity gathered by cantonal forestry offices. From a historical perspective, it has been recently assessed by Vieli (2017). However, quantitative data on the weather of this avalanche winter is restricted to sparse station observations so far. The gridded weather reconstructions presented in this paper can help to analyse 1887/88 winter weather quantitatively, thus helping to better understand weather patterns leading to such an event. For demonstration purposes, we performed some simple calculations of monthly averages, mean
snow precipitation and the zero-degree level that are summarised in Fig. 8.

Shown are post-processed precipitation reconstructions aggregated over one month (a–d) for winter 1887/88 and the recently published monthly precipitation reconstructions by MeteoSwiss (Isotta et al., 2019) (e–h). Both datasets reveal similar patterns of monthly precipitation, although regional differences occur. More deviations between the dataset can be observed in the amount of reconstructed precipitation. From both datasets, large precipitation sums can be determined in December 1887 on
the northern flank of the Alps and in the Jura mountains. January shows only little precipitation with highest values in the north-eastern Alps. February and March show extreme precipitation values in Ticino and the Gotthard region, in March also over the remaining part of Switzerland.

Daily reconstructions allow for going more into detail. For example, we can calculate the development of the zero-degree level from gridded temperatures taking the intercept of a linear regression between temperature and altitude (bottom of Fig.
8). Altitude data used here was aggregated from the SRTM 90 digital elevation dataset (Jarvis et al., 2008) to fit the resolution of reconstructions. Another value of interest is the intensity of snowfall precipitation. In Fig. 8 (bottom), the average snow precipitation per snowfall area is shown, calculated from post-processed reconstructions and assuming an estimated 1 °C threshold (Jennings et al., 2018), below which precipitation falls as snow. Grey shaded areas depict periods of high avalanche activity as reported by Coaz (1889).

Analogue reconstructions (red) and assimilated temperature data (blue) show similar values. The extreme precipitation event in December 1887 coincides with a high altitude of the zero-degree level, thus leaving a snow covered area restricted to higher altitudes. This event was followed by several cold episodes and low precipitation in January. The first avalanche period was dominated by low temperatures and intensive precipitation. During the second avalanche period, the reconstructed zero-degree level rises to approximately 600 m a.s.l. with almost no snow precipitation. March shows two periods of high temperatures and
intensive precipitation in the middle of the month and during the third avalanche period.





Reconstructed precipitation patterns, as well as the development of temperature are in line with the findings from documentary data (Coaz, 1889) that report strong snowfall during December, a dry January and again intensive precipitation during February and March, especially in the Southern Alps. While the first two avalanche periods are determined by low temperatures, Coaz describes a sharp rise of the zero-degree level to about 2000 m a.s.l. preceding the third period that led to a high

number of wet avalanches, which can well be seen in the reconstructions. However, during the second period, reconstructions show relatively low snow precipitation values, contesting the high avalanche activity. Nonetheless, avalanches are not only triggered by intensive precipitation. For example, the intensive snowfall period in December 1887 and the first two in March 1888 are not accompanied by more frequent avalanches. To analyse this, also other factors like temperature and wind as well as the composition of different snow layers play an important role and have to be assessed. A closer look at these periods would

probably reveal more about the processes that triggered or prevented avalanches.

From precipitation and temperature reconstructions, new insight on the avalanche winter of 1887/88 can be gained already with simple methods. Using more sophisticated snow models that also take into account evaporation and snow-melt, high-resolution daily spatial data of the snow cover could be established that may be able to further explain avalanche activity. This is but one example, what the new daily reconstructions of temperature and precipitation could be used for. Analogue

reconstructions have already been applied as input to numerical models, such as crop modelling (e.g. Flückiger et al., 2017) or hydrological modelling (e.g. Brönnimann et al., 2018), but the list could be extended. Many other phenomena, e.g., heat waves or droughts can be analysed spatially, and making use of the long timespan changes of climate and extreme events over time could be investigated.

## 5 Conclusions

As shown in this paper, the Analogue Resampling Method is a suitable and efficient approach for reconstructing daily precipitation and temperature fields from station observations. Using CAP7 weather types as a criterion for physical consistency and a set of observations from 68 weather stations, we could present a long-term, physically consistent, high-resolution spatial dataset of these meteorological parameters for Switzerland since 1864. The datasets are published at [DATABASE, DOI]. Analogue reconstructions for temperature and precipitation show good results, but experience difficulties in reproducing ver-

tical temperature gradients and show a general negative bias for precipitation arising mainly from underestimation of extreme events. Furthermore, analogue reconstructions reveal difficulties to correctly distinguish between wet and dry days. On average 17% of days were wrongly assigned. Temperature reconstructions could be considerably improved by assimilating station data using an ensemble Kalman fitting approach. Assimilated temperature fields show average error magnitudes of less than 1 °C and are nearly unbiased for the mean. The issue with vertical temperature gradients could be largely eliminated, although in

winter some problems remain that could probably be referred to difficulties of reconstructions to determine inversion heights. Precipitation data were post-processed with quantile mapping, adjusting the distributions of daily precipitation for each grid cell to obtain more accurate values. The mean bias could be successfully reduced, while a larger uncertainty for extreme events



persists. However, error values show a slight increase in post-processed data. With the simple approach of quantile mapping presented in this paper, the problem of wet and dry day discrimination could not be addressed.

There are some limitations to the analogue method, as the availability and coverage of station observations affects the accuracy of the results, especially for precipitation reconstruction. In regions with sparse information from weather stations, the uncertainty of reconstructions is larger. In Switzerland, this regards mostly mountainous areas. A second constraint is the comparatively small size of the analogue pool that is available for this application, which is especially relevant for extreme events as for such events less suitable analogues exist. To reconstruct extremes more accurately, notably for precipitation, a

longer series of spatial data and a denser station network would be needed. With more sophisticated post-processing methods for precipitation, also errors in wet and dry day discrimination could be reduced. As mentioned, the analogue approach does not guarantee temporal consistency and therefore isn't completely suitable to analyse trends. However, the dataset presented in this study very well complements the monthly reconstructions by Isotta et al. (2019) that were specifically designed for this purpose.

The assessment of avalanche winter 1887/88 in Switzerland shows that the reconstructed development of temperature and precipitation correspond well to documentary sources and to monthly reconstructions by Isotta et al. (2019). Possible applications of our daily, high-resolution precipitation and temperature reconstructions range from crop modelling to the reconstruction of river runoffs, and the study of weather phenomena in the context of climate change.

  Could daily reconstructions be extended even further back in time? For Switzerland, a recent survey brought to light a large

amount of early instrumental data (Pfister et al., 2019). An extension of the dataset to the pre-industrial period is therefore envisaged, although larger measurement errors and less consistent measurement series make this endeavour rather challenging. The method should also be suitable to reconstruct daily meteorological fields for other regions of Central and Western Europe.

*Data availability.* Reconstructed daily precipitation and temperature datasets over the period 01.01.1868–31.12.2017 are published at the open-access repository PANGAEA (https://doi.org/10.1594/PANGAEA.907579; Pfister, 2019) under a Creative Commons BY-NC-SA li-

cence.

*Author contributions.* LP designed the study and performed the analysis with support from SB, FI and PH. MS provided weather type data and FI monthly reconstructions used in section 4.4. CH contributed to the analysis of historical avalanche winter 1887/88.

*Competing interests.* The authors declare that they have no conflict of interest.

*Acknowledgements.* This work has been supported by Swiss National Science Foundation projects CHIMES (169676). Station and gridded

data of meteorological parameters were obtained by courtesy of MeteoSwiss. All methods were computed in R (R Core Team, 2017), using





the quantile mapping package from Gudmundsson et al. (2012) and in Python. We would like to thank all researchers that assisted with advice and input on methodological matters.





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





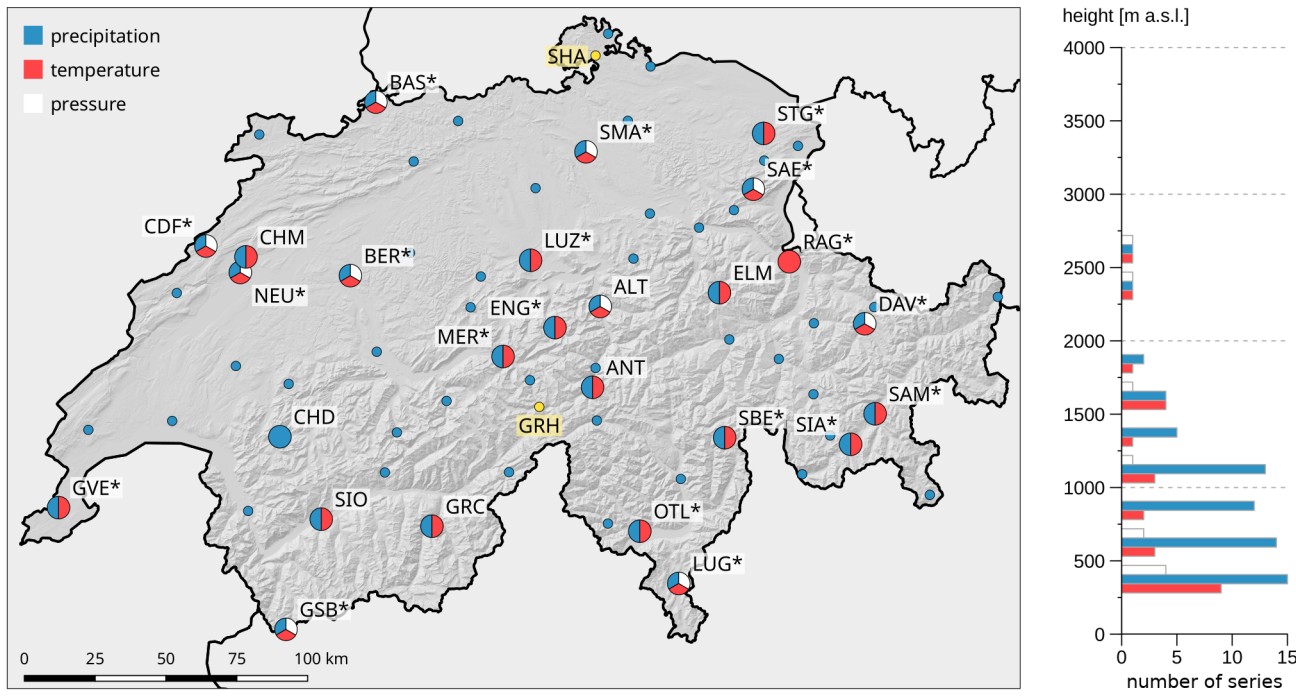

**Figure 1.** Station Map. (left) Measured variables are indicated as colours. Labelled pie charts represent NBCN climate monitoring stations. Additional NBCN precipitation stations are indicated as small blue dots. Stations that were used for temperature assimilation are marked by an asterisk. Yellow dots represent series used for station-based validation. (right) Vertical distribution of measurement series is indicated by altitude class for each variable.





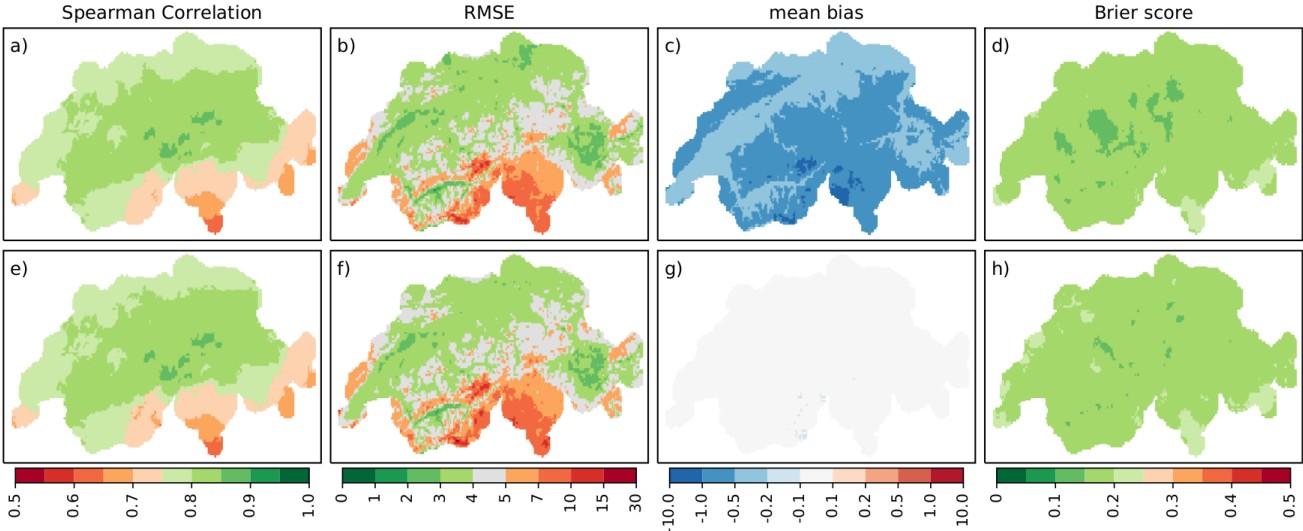

**Figure 2.** Validation over time of precipitation 1961–2017 for analogue reconstructions (a–d) and quantile-mapped data (e–h). Shown are Spearman correlation (a, e), RMSE in mm (b, f), mean bias in mm (c, g) and Brier score (d, h).





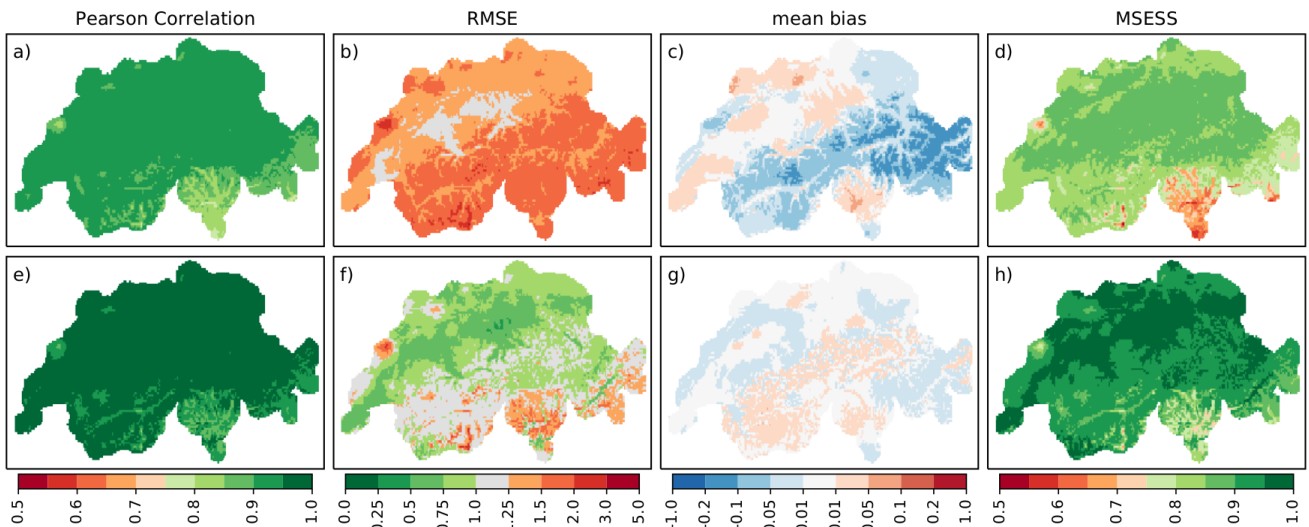

**Figure 3.** Validation of temperature over time 1961–2017 for ARM (a–d) and EnKF (e–h) reconstructions. Shown are Pearson correlation (a, e), RMSE in mm (b, f), mean bias in mm (c, g) and MSESS (d, h).





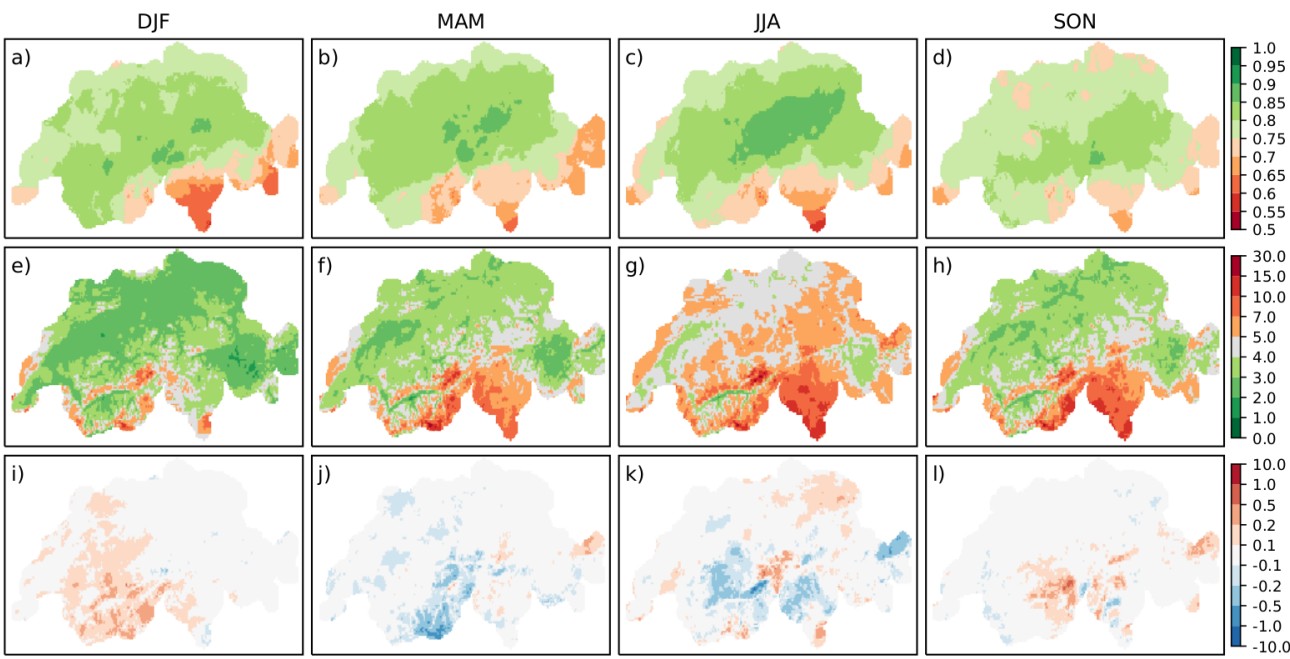

**Figure 4.** Validation of quantile-mapped Precipitation over time by season. Shown are Spearman correlation (a–d), RMSE (e–h) and bias (i–l).



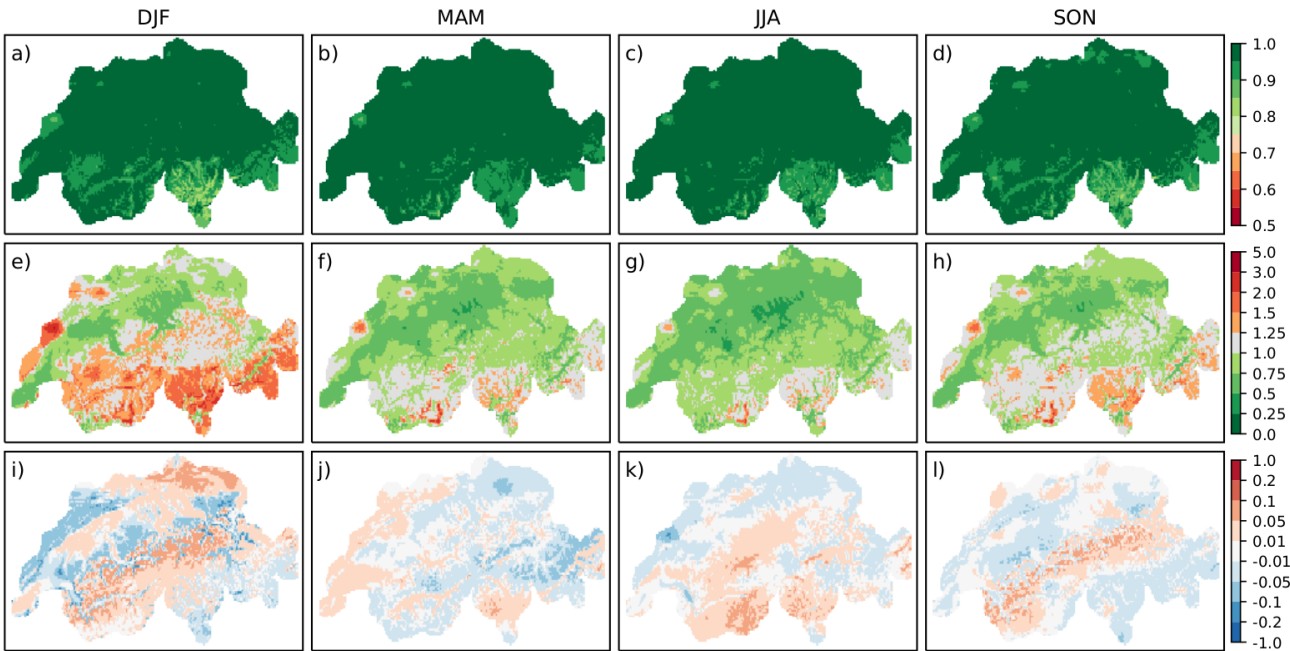

**Figure 5.** Results of validation over time of EnKF temperature reconstructions for each season. Shown are Pearson correlation (a–d), RMSE (e–h) and bias values (i–l).



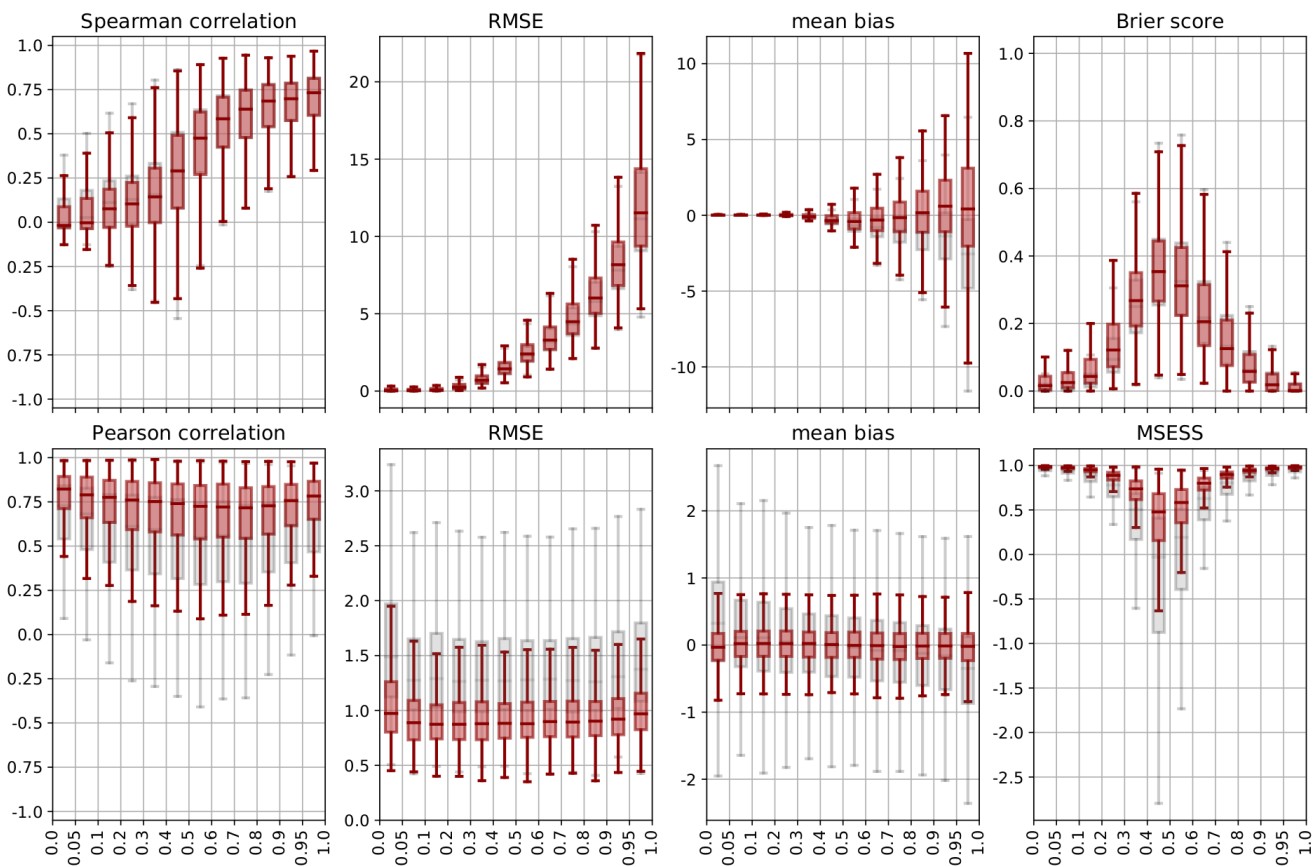

**Figure 6.** Validation over space of precipitation (top) and temperature (bottom) for ARM reconstructions (grey) and post-processed data (red), separated by quantile groups of spatial average precipitation and temperature, respectively. Shown are Spearman (precipitation) and Pearson (temperature) correlation, RMSE, bias, Brier score (precipitation) and MSESS (temperature). Boxes range from the 1st to the 3rd quartile and whiskers extend to 1.5 times the interquartile range outside the box.





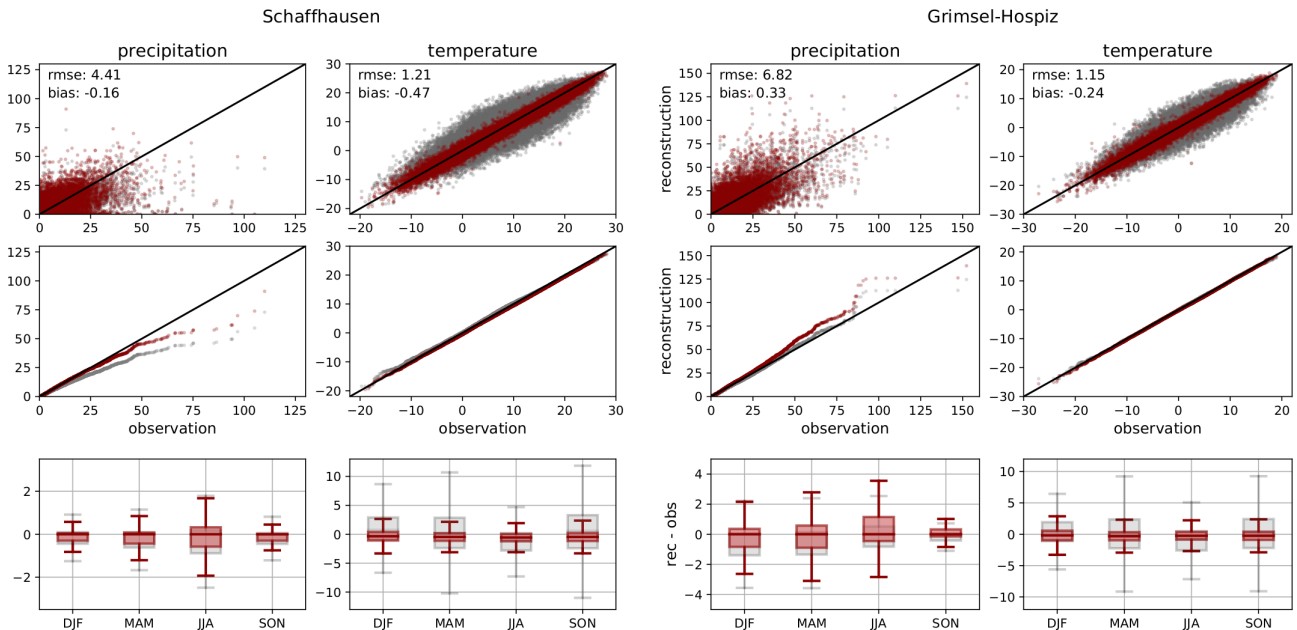

**Figure 7.** Comparison between reconstructions and station observations of precipitation [in mm] and temperature [in °C] for Schaffhausen (left) and Grimsel-Hospiz (right) with ARM reconstructions (grey) and post-processed data (red). Shown are observed vs. reconstructed values (top), quantile-quantile plots (centre) and boxplots of the deviation between reconstruction and observation by season (bottom).

**Figure 8.** Avalanche winter 1887/88: monthly mean precipitation from December 1887 to March 1888 calculated from post-processed daily reconstructions (top, a–d) compared to monthly reconstructions from Isotta et al. (2019) (top, e–h). On the bottom, estimated zero-degree level from ARM (red lines) and EnKF (blue lines) reconstructions are indicated, as well as average snow precipitation (blue bars), calculated from post-processed data. Grey shaded areas depict periods of increased avalanche activity as determined by Coaz (1889).