# Peer review of "Statistical Reconstruction of Daily Precipitation and Temperature Fields in Switzerland back to 1864"

_Climate of the Past, 2019_

## Referee Comment (RC1) · Anonymous Referee #1 · 19 Nov 2019

The manuscript presents a gridded reconstruction of daily precipitation and temperature in Switzerland over the last 140 years approximately, based on available but sparse station observations. The methodology is based on the Analogue Resampling method , with post-processing applied Quantile Mapping and Ensemble Kalman Filter. The study is indeed very interesting – the idea of combining the Kalman Filter and the analog method is I think novel. The applied methodology is valuable. The manuscript is generally clearly written and well structured. Therefore, I m happy to recommend the manuscript for publication after some revisions, which I hope that the authors may want to consider.

[Figure]

General comments:

1) The manuscript discusses at length the success and deficiencies of the reconstructions, both with the ARM and the post-processed reconstructions. This discussions is focused on the replication of the mean, variability and extreme events. I have one general comment in this regard. The ARM using just one analogue is in principle unbiased and should also replicate the correct variance, since it is simply a re-sampling of observations. Therefore, deficiencies in the replication of aggregated statistical measures, such as mean and variance, found in the same 'pool' period 1961-2017 can only be originated in the predictand field, the girdled temperature and precipitation products. (Of course, the skill in replicating the temporal succession and extremes is another question). Thus, the evaluation of the ARM by the leave-one-out method is actually not only a validation itself but also in combination with the gridded temperature and precipitation fields. Since the construction of these fields always involves some sort of regression or averaging of station data, the extremes and in general the variability is reduced compared to station data.

2) I understood why the station predictor data need to be de-seasonalized and standardized, as temperature and precipitation have different variation ranges. However, I did not understand why the gridded predictand fields also need to undergo this per-processing. In theory, once the ensemble of n analogues is identified, the same days can be selected from the pool of un-preprocessed predictand fields. Perhaps, the Kalman Filter algorithm requires that per-processing, but it is not obvious to me. A short explanation, if that is the case, would help the reader.

3) Through the manuscript, especially in the beginning I had problems to figure out which data are the 'predictors' and which the 'predictand'. It becomes clearer later in the manuscript, but perhaps the authors would like to use this terminology or a similar one from the start. It will help those readers that are not that acquainted with the analog method

Particular comments:

Some refer to the English usage, but I am not a native speaker, so the authors may want to double-check

line 3 ' whereas prior to that local station observations '

the sentence is ambiguous : whereas prior to that year, local stations observations..

line 37 'The analogue approach makes use of this statistical relationship between large-scale and local weather or meteorological patterns, while the former is used to predict the latter. '

what is 'the former' and which 'the latter' ?

Line 81: 'Errors are estimated to be in the order of factor 1.7 for precipitation on) and 1.3 for precipitation above the 90% quantile..

I guess units are mm/day

line 104 data to predict the spatial fields and a record of the spatial data from which the reconstructions are drawn. the spatial fields we used daily station observations, while the RhiresD and TabsD datasets for 1961–2017 from M

I would set here which are data are the predictors and which the predictands. Many readers would refresh their understanding of the method by going directly to this section

line 115 The day of interest and possible analogue days are required to be of the same WT to assure similar synoptic-scale

to ensure

line 158: where x denotes the updated state vector (analysis), x and y as described above and K is the Kalman gain or innovation matrix calculated from the ensemble. In this and the following equations, H describes the Jacobian matrix of H(x) and extracts

I am not sure this is the Jacobian matrix. In my undesrstanding the Jacobian of a

vector function of several variables is constructed by taking the partial derivatives along the vector dimensions. Here, I think the authors mean the projection operator or the selection operator

line 191 I think that QM becomes necessary because of the use of the Ensemble Kalman Filter. The ARM (best analog) would deliver the correct pdf (unbiased, correct stdev, etc). Also an ARM based on an ensemble of analogues would need QM. Is that true ? Perhaps the authors may want to discuss this point.

Line 338: From this, we can conclude, that reconstructions provide accurate precipitation fields for low to moderate precipitation

delete comma after conclude

line 340 'Extreme events, however, are underestimated by ARM reconstructions and show large errors also for post- processed data. As extreme events by definition occur more rarely, the number of suitable analogues is limited. As argued in for upper and lower extreme values. In general, errors could be significantly reduced with Kalman fitting. The average bias reveals, that while analogue reconstructions tend to overestimate negative extreme values and underestimate extremely high'

I am not sure that I completely agree (see previous comments). The ARM (best analogue) would automatically produce the correct pdf, It would miss extremes, and produce them at the worng point in time, but the pdf should be the best possible (it is simply a re-sampling of the observations). I agree that the Kalman filter, and in general an 'ensemble ARM' would produce smaller RMSE at the expense of reduced variability, but trade-off belongs to the general statistical trade-off between bias and variance of an estimator.
* * *

---

## Referee Comment (RC2) · Anonymous Referee #2 · 6 Dec 2019

The authors present a new dataset based on a combination of long station records and the analogue resampling method for daily temperature and precipitation. These fields are then adjusted using ensemble Kalman filtering or quantile matching. Both the non-adjusted as the adjusted dataset are validated using a leave-one-out approach and against independent station observations. Finally, an application of the dataset is given in a reconstruction of snowfall and the altitude of the 0 degree line to better understand a historic avalanche winter.

The study is sound and - as far as I can tell - no methodologic errors have been made. The study is a pleasure to read and the application, presented like it is the cherry on the

cake, makes a compelling case for the dataset. Although I am quite enthusiastic about this study, there are three aspects which the authors may want to look into. One is the need for some additional explanation, one relates to an issue with the post-processing and the last one relates to the analogue method and a suggestion to overcome the drawback of the limited number of suitable analogues.

1. On page 7 (line 189) is is argued that 'reconstructions are often affected by biases in the mean, an increased number in wet days and underestimation of extreme events'. This statement is corroborated by a reference to Piani et al. This study works with global climate model data and a global dataset of hydrological forcing data. It is common knowledge that such global datasets suffer from the problems described on line 189, but one of the appealing aspects of the analogue method is that it has the potential to avoid these 'smoothing' problems. After all, it are observed situations that are used to build the reconstruction (including observed extremes) rather than a watered-down statistical interpolation. A more clear view on WHAT the reason is that the ARM provides estimates that have too many wet days, lack real extremes and suffer from a bias. After all, much of the study is devoted to adjusting for these problems.

2. On page 8, line 213, the authors state that the assumption in the post-processing method is that the precipitation distribution is not subject to changes in time. The period the authors use to calcuate the parametric transfer functions is 1961-2017. Obviously, this period includes the climate change effectson the precipitation which are also evident in the Swiss climate. Examples of time series with steep trends and/or decadal variability of e.g. RR1 (number of wet days) are Andermatt and Altdorf, extreme precipitation has changed as well, as evident in e.g. R95p in Basel-Binnigen. Can the authors comment on how climate change and decadal variability affects the effectiveness of the adjustment for precipitation?

3. A problem with the analogue method, which the authors mention several times in the study, is the limited number of analogues. Earlier, Van den Dool (1994, his section 5) stumbled upon this problem as well and he suggests a way out. He suggests to construct an analogue having greater similarity than the best natural analogue. He considers linear combinations of naturally occuring analogues. There are a few differences between the Van den Dool study and the current study (monthly vs. daily fields for instance), but it may be worth looking into this suggestion as it may make the dataset presented in this study stronger.

Other (minor) things the authors may want to look into

- page 5, line 120. What is the motivation to set this window to 60 days (and not e.g. 90 of 30)?

- page 6, line 172, an observation error of 1C is quick steep - is there a sound reason for taking it that large?

very very minor remarks

- line 185, in my humble view, observations are not corrected but adjusted (as I think that an observation is not 'wrong')

- line 223, change 'chapter' to 'section'

- line 490, the family name of the 2nd author is 'van Leeuwen' and his initials are P.J.

- caption figure 8, in my print out, the snow precipitation bars are grey and the avalanche acitvity periods are brownish

10.5194/cp-2019-124
2019

Reference

Van den Dool, H. M. (1994). Searching for analogues, how long must we wait?. Tellus A, 46(3), 314-324.
* * *
RR1: Wet days (RR >= 1 mm)
1642 Andermatt, SWITZERLAND
ANNUAL

**Fig. 1.** Number of anual rainy days for Andermatt

R95p: Days with RR > 95th percentile of daily amounts (very wet days)
239 Basel Binningen, SWITZERLAND
ANNUAL

European Climate Assessment & Dataset  http://www.ecad.eu
Time series plot created on 06-12-2019

**Fig. 2.** Number of very wet days for Basel-Binningen

---

## Referee Comment (RC3) · Anonymous Referee #1 · 16 Dec 2019

After submitting my review, I stumbled upon a reprint that described a similar method, although not in climate context. The authors may want to refer to in in the revised version: The Analog Ensemble Kalman Filter and Smoother https://hal.archives-ouvertes.fr/hal-01188825

---

## Author Comment (AC1) · 29 Jan 2020

**Reply to the reviewers comments**

The manuscript presents a gridded reconstruction of daily precipitation and temperature in Switzerland over the last 140 years approximately, based on available but sparse station observations. The methodology is based on the Analogue Resampling method , with post-processing applied Quantile Mapping and Ensemble Kalman Filter. The study is indeed very interesting – the idea of combining the Kalman Filter and the analog method is I think novel. The applied methodology is valuable. The manuscript is generally clearly written and well structured. Therefore, I m happy to recommend the manuscript for publication after some revisions, which I hope that the authors may want to consider.

We'd like to thank the reviewer for this positive feedback and for all the helpful comments and suggestions to improve the manuscript.

**General comments:**
1) The manuscript discusses at length the success and deficiencies of the reconstructions, both with the ARM and the post-processed reconstructions. This discussions is focused on the replication of the mean, variability and extreme events. I have one general comment in this regard. The ARM using just one analogue is in principle unbiased and should also replicate the correct variance, since it is simply a re-sampling of observations. Therefore, deficiencies in the replication of aggregated statistical measures, such as mean and variance, found in the same 'pool' period 1961-2017 can only be originated in the predictand field, the girdled temperature and precipitation products. (Of course, the skill in replicating the temporal succession and extremes is another question). Thus, the evaluation of the ARM by the leave-one-out method is actually not only a validation itself but also in combination with the gridded temperature and precipitation fields. Since the construction of these fields always involves some sort of regression or averaging of station data, the extremes and in general the variability is reduced compared to station data.
This is an important point. We agree with the reviewer that in principle, the ARM using only the best analogue has the advantage over e.g. simple interpolation methods to reproduce natural variability and mean values. However, as the reviewer points out, certain methodological choices as the coupling of temperature and precipitation reconstructions or also the application of a distance measure over all station data can result in reduced variance and biases. The best analogue thus represents a best compromise to optimally satisfy all criteria described in section 3.1 of the manuscript. Some causes of reduced variance and bias (limited size of the analogue pool, availability of station data) are discussed in the manuscript. In section 3 of the revised manuscript, we will state more clearly that the ARM generally should be unbiased and reproducing natural variability and we will add a few sentences on possible impacts that methodological choices may have, limiting the capability of the ARM to do so.

2) I understood why the station predictor data need to be de-seasonalized and standardized, as temperature and precipitation have different variation ranges. However, I did not understand why the gridded predictand fields also need to undergo this preprocessing. In theory, once the ensemble of n analogues is identified, the same days can be selected from the pool of un-preprocessed predictand fields. Perhaps, the Kalman Filter algorithm requires that preprocessing, but it is not obvious to me. A short explanation, if that is the case, would help the reader.
Thank you for this remark. Station data are standardized and temperature measurements also de-seasonalised. To the gridded data, however, no standardisation is applied, but only a de-seasonalising of temperature fields. For precipitation, the best analogue dates are selected directly from the pool of un-processed predictand fields (absolute values). As analogues are calculated using temperature deviations from a mean seasonal cycle, reconstructed temperature fields are accordingly taken from pre-processed gridded data (temperature anomalies). The mean climatology

is then added again to get absolute temperature values. This procedure might not be entirely clear from the formulation in section 3.1 and we will try to clarify it in the revised manuscript.

3) Through the manuscript, especially in the beginning I had problems to figure out which data are the 'predictors' and which the 'predictand'. It becomes clearer later in the manuscript, but perhaps the authors would like to use this terminology or a similar one from the start. It will help those readers that are not that acquainted with the analog method
Thank you for this suggestion. For the analogue method, the spatial fields of a given day of interest is the predictand and all data used to look for the best analogue (station data, weather types) are used as predictors. In the revised version, we try to make this distinction clearer for better understanding.

Particular comments:

Some refer to the English usage, but I am not a native speaker, so the authors may want to double-check

Thank you for these comments. We changed the order of the reviewer's comments in the following to first answer remarks regarding language and then go into detail with comments regarding content and understanding of the manuscript.

line 3 ' whereas prior to that local station observations '
the sentence is ambiguous : whereas prior to that year, local stations observations..

line 115 The day of interest and possible analogue days are required to be of the same WT to assure similar synoptic-scale
to ensure

Line 338: From this, we can conclude, that reconstructions provide accurate precipitation fields for low to moderate precipitation
delete comma after conclude

Thank you for these suggestions. We will correct the errors and adjust the wording in the manuscript for better understanding.

line 37 'The analogue approach makes use of this statistical relationship between large-scale and local weather or meteorological patterns, while the former is used to predict the latter. '
what is 'the former' and which 'the latter' ?
This is a good point. In fact, the analogue method can be applied in both ways: for downscaling large-scale weather data to a local scale, as well as to predict large scale weather data from local scale information. As in the introduction we want to keep the description of the method general, the wording will be changed in the revised manuscript as follows: "The analogue approach makes use of this statistical relationship between large-scale and local weather or meteorological patterns, while one can be used to predict the other". Further details are given in section 3.

Line 81: 'Errors are estimated to be in the order of factor 1.7 for precipitation on) and 1.3 for precipitation above the 90% quantile.
I guess units are mm/day
Thank you for this comment. In the description of the RhiresD dataset (MeteoSwiss, 2016a), standard errors of the dataset compared to local point observations are indicated to be in the order of a factor between 1.3 to 1.7 (dimensionless).

line 104 data to predict the spatial fields and a record of the spatial data from which the reconstructions are drawn. the spatial fields we used daily station observations, while the RhiresD and TabsD datasets for 1961–2017 from M

I would set here which are data are the predictors and which the predictands. Many readers would refresh their understanding of the method by going directly to this section

Thank you. In accordance with comment 3), we will clarify the terminology in the revised manuscript.

line 158: where x denotes the updated state vector (analysis), x and y as described above and K is the Kalman gain or innovation matrix calculated from the ensemble. In this and the following equations, H describes the Jacobian matrix of H(x) and extracts

I am not sure this is the Jacobian matrix. In my undesrstanding the Jacobian of a vector function of several variables is constructed by taking the partial derivatives along the vector dimensions. Here, I think the authors mean the projection operator or the selection operator

Generally H(x) is the symbol for the operator and H is the Jacobi matrix of the H(x), so Jacobi matrix would be correct. However, in this particular case, H describes indeed a simple selection operator, as no transformation of the data is done.

line 191 I think that QM becomes necessary because of the use of the Ensemble Kalman Filter. The ARM (best analog) would deliver the correct pdf (unbiased, correct stdev, etc). Also an ARM based on an ensemble of analogues would need QM. Is that true ? Perhaps the authors may want to discuss this point.

Thank you. The Ensemble Kalman Filter is only applied as post-processing of temperature fields, but was discarded in favor of the simpler method of quantile mapping for precipitation. The necessity of post-processing of precipitation is mainly related on the reasons discussed in the answer to the reviewer's comment 1).

line 340 'Extreme events, however, are underestimated by ARM reconstructions and show large errors also for post- processed data. As extreme events by definition occur more rarely, the number of suitable analogues is limited. As argued in for upper and lower extreme values. In general, errors could be significantly reduced with Kalman fitting. The average bias reveals, that while analogue reconstructions tend to overestimate negative extreme values and underestimate extremely high'

I am not sure that I completely agree (see previous comments). The ARM (best analogue) would automatically produce the correct pdf, It would miss extremes, and produce them at the wrong point in time, but the pdf should be the best possible (it is simply a re-sampling of the observations). I agree that the Kalman filter, and in general an 'ensemble ARM' would produce smaller RMSE at the expense of reduced variability, but trade-off belongs to the general statistical trade-off between bias and variance of an estimator.

Thanks for this point. Referring to the authors' response to comment 1), there are certainly various reasons that limit the capability of the ARM to correctly reproduce the correct pdf, which will be discussed in more detail in section 3. As shown in figure 6 (over space) and figure 7 (for stations over time) such deviations between ARM reconstructions and observed distributions exist and can be associated with upper (and for temperature also lower) extreme values. Lines 340ff (figure 6) refer to the validation over space; in this particular case, also uncertainties originating from a sparse station coverage play an important role. While the ARM assuming an unlimited pool of possible analogues would produce a correct pdf over time, uncertainties regarding spatial patterns in regions without measurements would still persist.

---

## Author Comment (AC2) · 29 Jan 2020

**Reply to the reviewers comments**

The authors present a new dataset based on a combination of long station records and the analogue resampling method for daily temperature and precipitation. These fields are then adjusted using ensemble Kalman filtering or quantile matching. Both the non-adjusted as the adjusted dataset are validated using a leave-one-out approach and against independent station observations. Finally, an application of the dataset is given in a reconstruction of snowfall and the altitude of the 0 degree line to better understand a historic avalanche winter.

The study is sound and - as far as I can tell - no methodologic errors have been made. The study is a pleasure to read and the application, presented like it is the cherry on the cake, makes a compelling case for the dataset. Although I am quite enthusiastic about this study, there are three aspects which the authors may want to look into. One is the need for some additional explanation, one relates to an issue with the post-processing and the last one relates to the analogue method and a suggestion to overcome the drawback of the limited number of suitable analogues

We thank the reviewer for the very positive feedback and appreciate the valuable suggestions and comments.

1. On page 7 (line 189) is is argued that 'reconstructions are often affected by biases in the mean, an increased number in wet days and underestimation of extreme events'. This statement is corroborated by a reference to Piani et al. This study works with global climate model data and a global dataset of hydrological forcing data. It is common knowledge that such global datasets suffer from the problems described on line 189, but one of the appealing aspects of the analogue method is that it has the potential to avoid these 'smoothing' problems. After all, it are observed situations that are used to build the reconstruction (including observed extremes) rather than a watered-down statistical interpolation. A more clear view on WHAT the reason is that the ARM provides estimates that have too many wet days, lack real extremes and suffer from a bias. After all, much of the study is devoted to adjusting for these problems.
This is an excellent suggestion. As the reviewer states, the analogue method generally has the advantage over e.g. statistical interpolation to reproduce natural variability and mean values. However, given the assumptions made in the setup of the method (e.g. similarity criterion, coupled reconstruction of temperature and precipitation) and limitations of available data (e.g. size of analogue pool, coverage of station data), also analogue reconstructions can suffer from the problems described in line 189. In the revised manuscript, we will state that more clearly and go further into detail about possible consequences of methodological choices on resulting reconstructions in section 3.1 (see also reply to reviewer's comment RC1) and adopt this argumentation in section 3.2 instead of the mentioned reference to literature. In section 4, the limited size of the analogue pool and relatively sparse station coverage are identified as the main causes for problems regarding the reconstruction of extreme events and the related bias in the mean. Tests for the period 1961-2017 revealed that the size of the analogue pool is limited by the restrictions of the analogue method (seasonal window, weather types) to 1772 on average, with 21% of the days having less than 1000 and about 1% less than 500 possible analogues. Whereas for problems regarding the discrimination between wet and dry days, no detailed assessment has been carried out in order to limit the scope of the manuscript. It could be shown however, that the analogue method is prone to such problems and that especially for moderate precipitation events it fails to correctly reproduce precipitation areas (figure 6).

2. On page 8, line 213, the authors state that the assumption in the post-processing method is that the precipitation distribution is not subject to changes in time. The period the authors use to calculate the parametric transfer functions is 1961-2017. Obviously, this period includes the climate change effects on the precipitation which are also evident in the Swiss climate. Examples of time

series with steep trends and/or decadal variability of e.g. RR1 (number of wet days) are Andermatt and Altdorf, extreme precipitation has changed as well, as evident in e.g. R95p in Basel-Binnigen. Can the authors comment on how climate change and decadal variability affects the effectiveness of the adjustment for precipitation?

Thank you for this important question. While the effects of climate change and decadal variability on precipitation are captured by the analogue method to the extent where they can be found in station data or changes in the occurrence of weather types, post-processing does not take such effects into account. This very simple setting of quantile mapping was chosen to avoid over-fitting to the period 1961-2017, as the correction is applied to the whole dataset back to 1864. However, whether adjustments by quantile mapping show a pattern that can be related to climatic changes or decadal variability has not been analysed. Nonetheless, as quantile mapping does not correct the number of wet days and an increase in the number of extreme events related to climate change is already captured by the analogue method (from station data and weather types), the impact of climate change or decadal variability on the effectiveness of the chosen post-processing approach is limited to the intensity of extreme precipitation. Considering the large uncertainties in the reconstruction of extremes compared to the magnitude of corrections by quantile mapping (see e.g. figure 7), the adjustment can be considered very effective albeit being calibrated for a period subject to climatic changes.

3. A problem with the analogue method, which the authors mention several times in the study, is the limited number of analogues. Earlier, Van den Dool (1994, his section 5) stumbled upon this problem as well and he suggests a way out. He suggests to construct an analogue having greater similarity than the best natural analogue. He considers linear combinations of naturally occuring analogues. There are a few differences between the Van den Dool study and the current study (monthly vs. daily fields for instance), but it may be worth looking into this suggestion as it may make the dataset presented in this study stronger.

Thank you for pointing out this interesting approach by Van den Dool. As our study has the advantage to dispose of a much larger pool of analogues and to cover a smaller area of study than the Van den Dool study, it is easier to find better matching analogues. Together with post-processing, reconstructions show very satisfying results. Nonetheless, it would be worth examining Van den Dool's method for the reconstruction of daily precipitation and temperature fields in Switzerland and we will definitely consider this suggestion for future work.

We will add the following sentence to the conclusions in the revised manuscript: "Another option to address the problem of small analogue pools as proposed by Van den Dool (1994) is to construct more similar analogues by linear combination of several possible analogue dates."

Other (minor) things the authors may want to look into
• page 5, line 120. What is the motivation to set this window to 60 days (and not e.g. 90 of 30)?

For the analogue method, we tested different seasonal windows. In order not to constrain the analogue pool too much but still to have reconstructions with similar seasonal patterns, an optimum was found at about ±60 days. This value is also in line with literature (e.g. Horton et al., 2017; Caillouet et al., 2019; Ben Daoud et al., 2016; all cited in the manuscript).

• page 6, line 172, an observation error of 1C is quick steep - is there a sound reason for taking it that large?

As station measurements can be affected by micro-climatic conditions that are not captured by gridded data and due to larger uncertainties of the earlier observations, a rather conservative observation error of 1°C was chosen.

very very minor remarks
• line 185, in my humble view, observations are not corrected but adjusted (as I think that an observation is not 'wrong')

• line 223, change 'chapter' to 'section'
• line 490, the family name of the 2nd author is 'van Leeuwen' and his initials are P.J.
• caption figure 8, in my print out, the snow precipitation bars are grey and the avalanche acitvity periods are brownish

Thank you very much for these remarks; we will adjust them accordingly in the manuscript.
As for the colors in figure 8: they seem to be matching the description on screen and in my printout.

Reference
Van den Dool, H. M. (1994). Searching for analogues, how long must we wait?. Tellus A, 46(3), 314-324.

---

## Author Comment (AC3) · 29 Jan 2020

**Reply to the reviewers comments**

After submitting my review, I stumbled upon a reprint that described a similar method, although not in climate context. The authors may want to refer to in in the revised version: The Analog Ensemble Kalman Filter and Smoother https://hal.archives-ouvertes.fr/hal-01188825

We'd like to thank the reviewer for this addition. In the revised manuscript, this and a further article on this topic will be referred to in the introduction. We will add the following sentence in the revised manuscript: "The combination of the analogue method with a Kalman filter was tested e.g. by Tandeo et al. (2014) and Lguensat et al. (2017) for Lorenz models and has proven to provide good results."